# Earth Observation as a Facilitator of Climate Change Education in Schools: The Teachers' Perspectives

Panagiota Asimakopoulou *, Panagiotis Nastos [ID], Emmanuel Vassilakis [ID], Maria Hatzaki [ID] and Assimina Antonarakou [ID]

Department of Geology and Geoenvironment, National and Kapodistrian University of Athens, 10679 Athens, Greece; nastos@geol.uoa.gr (P.N.); evasilak@geol.uoa.gr (E.V.); marhat@geol.uoa.gr (M.H.); aantonar@geol.uoa.gr (A.A.)
* Correspondence: passimak@geol.uoa.gr

**Abstract:** Climate change education (CCE) fosters the skills and behavioral patterns of students in regards to climate-related challenges and risks. Despite its importance, the integration of CCE in schools is challenging due to the interdisciplinary nature of climate science and the obstacles and demands of everyday school reality. Here, we examine the case of satellite Remote Sensing (RS) for Earth Observation (EO) as an innovative tool for facilitating CCE. We focus on Greece, a country that, despite being a hot spot for climate change, shows a low level of CCE integration in schools and awareness for EO-based educational resources. Based on interviews with in-service teachers, our research reveals the following: (a) there is a high interest in how satellites depict environmental phenomena; (b) EO is considered an efficient vehicle for promoting CCE in schools because it illustrates climate change impacts most effectively; (c) local natural disasters, such as intense forest fires and floods, are more familiar to students and, thus, preferable for teaching when compared to global issues, such as the greenhouse effect and sea level rise; and (d) educators are in favor of short, hands-on, EO-based activities (also known as "activity-shots"), as the most useful material format for integrating climate change topics in their everyday teaching practice.

**Keywords:** remote sensing; school education; environmental education; classroom resources; Copernicus; ESA education

## 1. Introduction

Human sustainable development would not be feasible without a healthy planet [1]; thus, humanity urgently needs to implement the UN's Sustainable Development Goals (SDGs) [2] to combat climate change, conserve our oceans, and protect our forests (among other SDGs). In this context, climate change education (CCE) is critical for fostering the skills, attitudes, and behavioral patterns of young generations, to lead to a safer, greener, and fairer planet for all [3]. As such, the importance of CCE has been widely acknowledged and recognized by the international community. The United Nations Framework Convention on Climate Change [4], the Paris Agreement [5], and the European Green Deal [6] all consider schools as places to engage pupils on the changes needed to respond to urgent climate challenges. In response, the European Commission recently launched the Education for Climate Coalition [7] initiative, aiming to mobilize the education community towards a true bottom-up CCE engagement.

Despite the importance of CCE for sustainable development, the response from national education systems worldwide, in promoting relevant initiatives and programs in schools is considered insufficient [8]. At the national level, the effort of integrating CCE into the classroom is left to the teachers, who are faced with the novel challenge of having to teach their students a topic they never learned in their own schooling. Moreover, climate change is a complex subject because of its interdisciplinary nature, which covers a wide spectrum of different scientific domains, leading to confusion and misconceptions for

teachers (see for example [9–13]). In addition to the scientific complexity, climate change has also been a controversial topic, especially in parts of the world where their political environments, in previous years, were hostile toward environmental issues [9,14]. As such, CCE still puzzles educators today on how to constructively include it in their curriculums; it remains largely unclear what are the most effective teaching approaches and practices educators should use.

In order to identify the most effective teaching approaches for CCE, Monroe et al. [15] reviewed 49 studies assessing CCE interventions. Two common teaching strategies were identified among the successful educational programs within this sample:

1.   Making climate change information personally relevant and meaningful for learners.
2.   Using active and engaging teaching methods.

The use of satellite RS data and tools is additionally recognized as an effective teaching strategy for CCE in the same review [15]. Satellite RS is a powerful tool for monitoring global climate variables from space, offering a systematic coverage of the earth's surface and atmosphere at different spectral, spatial, and temporal resolutions. The RS data provided by EO satellites are being used for monitoring atmospheric composition, meteorological variables, land use changes, deforestation and desertification processes, ice sheets, glacier movements, sea surface levels and phytoplankton growth, among others. The satellite datasets span over a period of almost half a century, providing important information on the diachronic evolution of climate variables and potential trends in support of CC studies [16]. For example, recent climate studies using satellite RS data are already capable of proving direct evidence that anthropogenic activity has affected Earth's energy budget in the recent past, thus, accelerating CC [17].

As such, satellite RS for EO (from this point on, referred to only as EO), not only carries the native ability to visualize and depict CC causes, impacts, and trends, it also provides a unique source of information for CCE, and facilitates, at the same time, the implementation of the teaching strategies suggested by Monroe et al. [15]. Specifically, EO enables learners to focus on local CC impacts and, thus, makes CC personally relevant and meaningful. It also helps them understand difficult aspects of CC science (such as the remoteness of impacts, and the time lags between anthropogenic interventions and corresponding impacts to the climate system). Moreover, EO enables students to interact with authentic scientific data for studying changes taking place at multiple temporal and spatial scales [18,19]. As such, EO can be considered an innovative teaching and learning approach for CCE, bringing many benefits towards facilitating climate literacy. Additional EO benefits for CCE, according to a number of studies [20–25], include: (a) student motivation due to the attractiveness and fascination that space offers (e.g., students are drawn to satellite observations; thus, students are more engaged in the classroom). (b) The enhanced visualization of environmental problems, offering a high degree of vividness; the connection of school-derived theoretical knowledge with real world, up-to-date environmental issues. (c) Improvement of spatial orientation skills; enhancement of student critical thinking and decision-making competence. (d) The opportunity to implement a problem-solving oriented teaching approach, thus adopting a constructivist learning theory. (e) The opportunity for teachers to effectively engage their students with cutting-edge scientific discovery and technological innovation.

Despite the well-documented and recognized added value of EO for CCE, the level of EO utilization in schools varies in different countries for different reasons. In countries where efforts started earlier (such as the U.S., Germany, France, UK, China, and Italy) [23–28], EO uptake is still considered insufficient, in spite of the number of educational programs, activities, and tools developed for the respective educational systems [23–28]. This is mainly attributed to factors, such as (a) the limited supply of information suitable for non-experts; (b) the limited supply of regionally relevant case studies; (c) the emphasis on the remote sensing technology behind EO, which subsequently creates the need for prior teacher and student theoretical training [23–28]; (d) the need for specialized software and advanced technical infrastructure in school classrooms; and (e) the increased preparation-time for an EO-

based lesson [26,27]. On the positive side, the experiences gained in these countries over the last decade, coupled with continuous and rapid evolution of EO data and services, alleviated quite a few of the previous obstacles and complexities and, thus, created new opportunities for EO utilization in schools. Focusing specifically at the European level, the open data policy introduced by the Copernicus programs [29] not only offers satellite services for free, but also simplifies the process of accessing the data with free tools, such as the EO Browser [30] and the Sentinel Playground [31]. These recent developments significantly limit both the infrastructure requirements as well as the need for prior teacher knowledge and increased lesson preparation-time. With these intuitive and user-friendly tools, teachers are now capable of retrieving simple, true color images that directly illustrate many CC phenomena. Additional training could further enable teachers to unlock more information hidden in digital images. In this direction, the European Space Agency (ESA) is continuously developing and freely offering environmental EO-based educational activities for use during school hours (among other innovative classroom resources based on space themes) [32] as well as training opportunities to teachers of all educational levels [33]. ESA also addresses the language barriers by translating their classroom resources to the main European languages, as well as setting up Education Resource Offices (ESERO) [34] in each country in order to locally adapt materials according to national curricula, and provide training opportunities for teachers.

In our work, we examine the case of Greece, a country that, although located in a region most vulnerable to climate change [35], lacks a clear national mandate for integrating CCE in schools, unlike its neighboring country, Italy [36]. Moreover, Greece has not been efficient in utilizing modern technologies and tools, such as EO for facilitating CCE. This may be related to the fact that Greece does not hold an ESA ESERO Office, and only recently established its Hellenic Space Center for coordinating national space activities [37]. As such, Greece shows a relatively low level of EO awareness and usage in formal education [22,38], low availability of EO-based environmental education material in the Greek language, and a low level of coordination towards promoting CCE through EO, despite some notable, but fragmented academic initiatives taken in the past [38,39]. Here, we examined these issues by conducting a research survey with in-service primary and secondary public-school teachers. The main aim of the research is to explore the existing awareness and usage of EO data in Greek schools, measure the level of interest for EO utilization in CCE, as well as gather teacher needs for future development of environmental EO-based educational material. Although this is a nationally contextualized study, we expect that our research findings provide useful input to a wider audience interested in utilizing EO for CCE.

In Section 2, we describe the pre-research phase of our study and the methodology selected for the main research that followed. Then, we present the profiles of our study participants in Section 3.1 and, subsequently, the research findings in Section 3.2. Our results are discussed in Section 4, along with our conclusions and key recommendations for future actions towards an enhanced integration of EO in CCE.

## 2. Materials and Methods

A combination of survey techniques was used, aiming to explore the level of teacher awareness, interest, needs, and preferences for the use of EO in CCE. First, during the pre-research phase, we conducted unstructured interviews with eight (8) primary and secondary school teachers. The aim of this phase was to identify the key questions that would formulate an appropriate questionnaire for the main survey to follow. The interviews were conducted with teachers attending the public awareness activities for EO, organized by the Laboratory of Climatology and Atmospheric Environment (LACAE) of the National and Kapodistrian University of Athens (NKUA) in the framework of its participation in the Copernicus Academy Network [40]. As part of this initiative, EO was introduced to schools across Greece, as well as science festivals, through interactive presentations and hands-on activities [41].

From this pre-research phase, it was apparent that quantitative research based on anonymous questionnaires would possibly be confusing (and likely repelling) for the participants. Even though such research would provide measurable results, there was a risk for a small response rate from the teaching community, which, according to the unstructured interviews, considers EO complex, difficult, and high tech. Therefore, for the main-research phase of our study, we decided to use semi-structured interviews (SSIs), considering that this methodology would better serve our objectives. This is because the SSIs facilitate understanding of complex or little-known issues through open-ended questions and extended probing, providing, at the same time, the necessary flexibility on the logical order of questions and allowing for new ideas and unforeseen aspects to emerge [42,43]. For this purpose, and in compliance with the research and literature objectives, an interview guide was prepared in the form of a semi-standardized questionnaire (Table 1), containing a blend of closed- and open-ended questions, as well as why and how follow-up questions [42,43].

**Table 1.** Overview of the interview guide structure used to record participants' responses.

| Section No. | Section Content |
|:---:|:---:|
| 1 | Introduction of the survey's objectives and scope (frequently presented verbally by the interviewer). |
| 2 | Interviewee profiling questions. |
| 3 | Questions for recording the level of didactic engagement with CCE topics in the teaching process. |
| 4 | Awareness, prior knowledge, and experience with EO material, and resources, questions. |
| 5 | Questions for recording the level of interest on the potential utilization of EO data in CCE. |
| 6 | Questions on teacher preferences, regarding the thematic content of the future EO-based educational material for CCE. |
| 7 | Questions on teachers' requirements regarding the format of future EO-based educational material for CCE. |

The interview guide specifically aimed to:

- Record teachers' awareness, prior knowledge, and possible experience with EO information and tools;
- Identify teachers' levels of interest in incorporating EO-based material in their teaching of climate change;
- Map EO thematic areas to school subjects for future educational material development;
- Catalogue teacher needs, requirements, and preferences on how to best integrate EO in CCE;
- Collect new ideas and allow unforeseen aspects to emerge (gain quality information).

The semi-structured interviews were conducted from December 2018 to May 2019. The interviews took place at three (3) museums of the School of Science of NKUA—the Mineralogy and Petrology Museum [44], the Paleontology Museum [45], and the Zoological Museum [46]. These museums attract visiting schools daily from the Athens metropolitan area and beyond. The museums provided us with the pool of participants for our research, but also with a convenient setting as far as space and time was concerned. Specifically, while visiting students were at the hands of the museum personnel, getting a guided tour, escorting teachers were free and available for a friendly conversation on the subject at hand, with plenty of time to devote.

The sample of participants was selected from the pool of escorting teachers, based on whether they taught STEM (Science, Technology, Engineering, and Mathematics) curricular subjects or whether they have—or intended to, in the future—engaged in CCE activities with their students. As a result, some of the escorting educators teaching physical educa-

tion, religious studies, etc., as well as some primary school teachers, systematically teaching only 1st, 2nd, and 3rd grade students, were self-excluded. The resulting total number of participants to our study was 84. The semi-structured interviews were conducted conversationally, face-to-face, with one respondent at a time. Interview duration ranged between forty minutes to an hour depending on teacher interest [41] (p. 493). Teacher interviews were not recorded electronically due to noise constraints and interruptions from students. Instead, detailed notes of participant responses were taken on the printed questionnaires, which included enough blank space for the open-ended questions. However, if answers were extensive, separate notes were also taken. Special care was given to particularly valuable or memorable comments, which were written down word-for-word in quotation marks and approved by the participants [41].

## 3. Results

### 3.1. Profiles of the Study's Participants

We start with the presentation of the demographic characteristics of the teachers interviewed, including their education, training background and years of experience, all serving as an indication to their profile. Of the 84 participants in total, 57 were primary school teachers (teaching students between the ages 6 to 12, i.e., first through sixth grade students) and 27 lower secondary school (LSS) teachers (teaching students between the ages 12 to 15, i.e., 7th through 9th grade students). We keep this distinction between the main levels of compulsory education (primary and lower secondary) throughout this section because of the major difference in their vocational training. Primary school teachers are not specialized in science teaching while LSS teachers are, although only in the science field of their expertise.

Regarding the gender characteristics of our sample, 64 were female (76%) and 20 were male (24%). However, when it came to primary school teachers, 49 were female (86%) and 8 were male (14%). For LSS teachers, 15 were female (56%) and 12 were male (44%). There was, therefore, a higher percentage of females in the primary education level participating in our study, compared to the lower secondary education level. This type of difference is similar to the one revealed by the gender indicators reported by the Organization for Economic Co-operation and Development (OECD) for Greece for 2018 [47].

In terms of age for the whole sample, only one (1) participant was under the age of 30 (1%) and one (1) over the age of 60 (1%). For the decades in between, 34 participants were between 41 and 50 years old (40%), 33 participants were between 51 and 60 years old (39%), and 15 participants were between 31 and 40 years old (18%) (see also Figure 1).

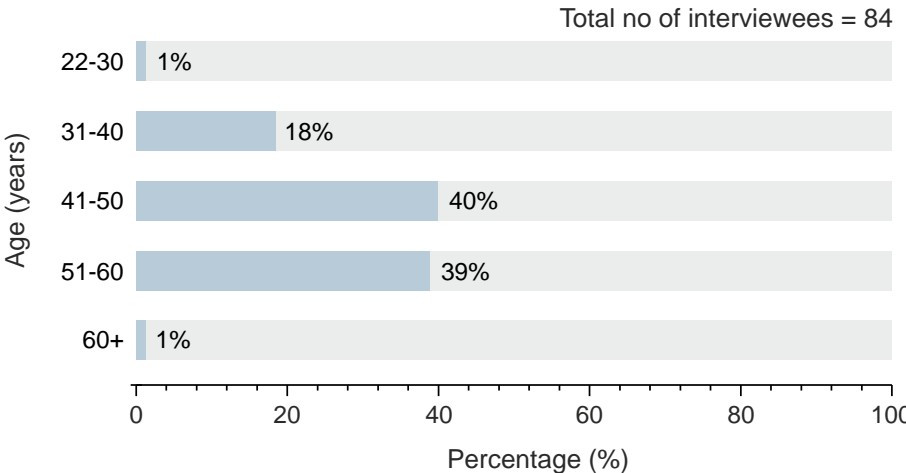

**Figure 1.** Age composition of teachers interviewed (percentages do not sum to 100 due to rounding).

However, if we look at the age distribution differences between the two levels of formal education, primary school teachers are found to be significantly older than LSS

teachers (in primary education, 46% of teachers are over 50 years old, while in lower secondary only 30% of teachers are over the age of 50). Accordingly, the level of teaching experience shows that only 1 teacher had between 0 and 4 years of experience (1%), 7 teachers between 5 and 10 (8%), 37 teachers between 11 and 20 (44%), 33 teachers between 21 and 30 (39%), and 6 teachers with over 30 years of experience (7%) (see also Figure 2). As such, the experience deviations between primary and lower secondary educators follow the age differences between the two levels of education.

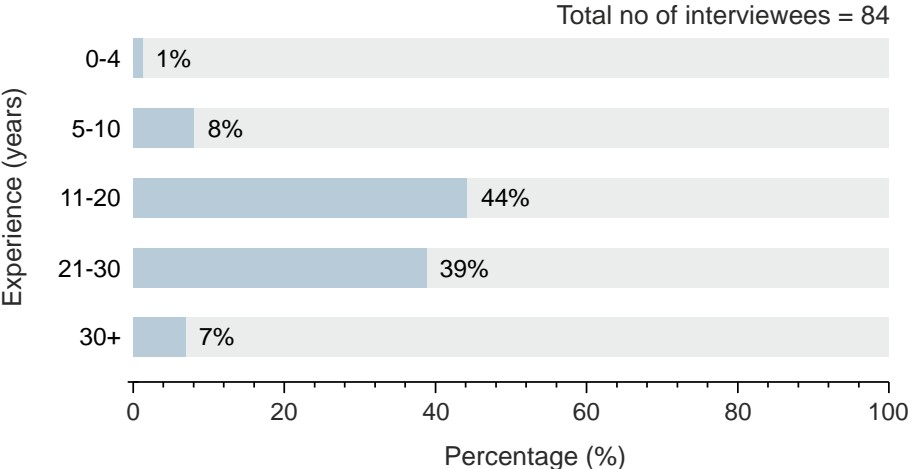

**Figure 2.** Years of participants' experience in state-run public schools (percentages do not sum to 100 due to rounding).

The analysis of Figures 1 and 2 reveal a highly experienced, although aged, group of participants in the current study. This finding coincides with the published OECD age indicators of teachers for 2018 [47], which show an aged teaching workforce in Greek schools compared to most European countries. This is probably due to the economic crisis and the austerity measures adopted by Greek governments during the last 10 years (since 2009), which included a total cut down of tenured teacher positions in public schools.

In regards to the participants' level of studies, 24 teachers (29%) had postgraduate studies (23 teachers has an MSc diploma, and 1 a PhD), while the majority, 60 teachers (71%), held a bachelor's degree in their field of teaching, as required by the state. However, if we look at the differences on the level of studies between primary and LSS teachers, we find that a much greater percentage of LSS teachers (44%) had postgraduate studies (MSc and PhD) compared to 21% of primary school teachers (see Figure 3). It should be noted here that, even at the first academic degree level, 4% of primary school teachers held a Pedagogic Academy diploma (see Appendix A.1.), which was a type of an associated degree, abolished by 1988 and replaced with the bachelor's degrees offered by the newly established Pedagogical Departments in Greek universities. Consequently, from 1989 onwards, there have been two pools of primary teachers eligible for recruitment by the state.

The final profile characteristic examined was teachers' participation in training programs for the utilization of digital technologies in the teaching practice. Specifically, we focused on the program called "In-Service Teacher Training in the Use of Information and Communication Technologies (ICTs) in Education", which was implemented over many years (2002–2020) in Greece (see Appendix A.2.). It trained thousands of teachers on A-level basic ICT skills and B-level advanced use of ICT in the teaching practice. According to the results of our study, out of the 84 teachers, the majority of 53 (63%) had successfully completed the A-level ICT training program, and out of those, 29 (35%) continued with the B-level phase of the program. Quite a significant number—31 teachers (37%)—never participated in any of these programs. Teachers abstaining from these training programs were fewer at the primary school level compared to the lower secondary level (32% against 48%, respectively). This difference can be attributed to, most likely, the ICT skill deficits of

the primary school teachers, possibly due to older age and lower level of studies, which possibly motivated them to compensate through these training programs.

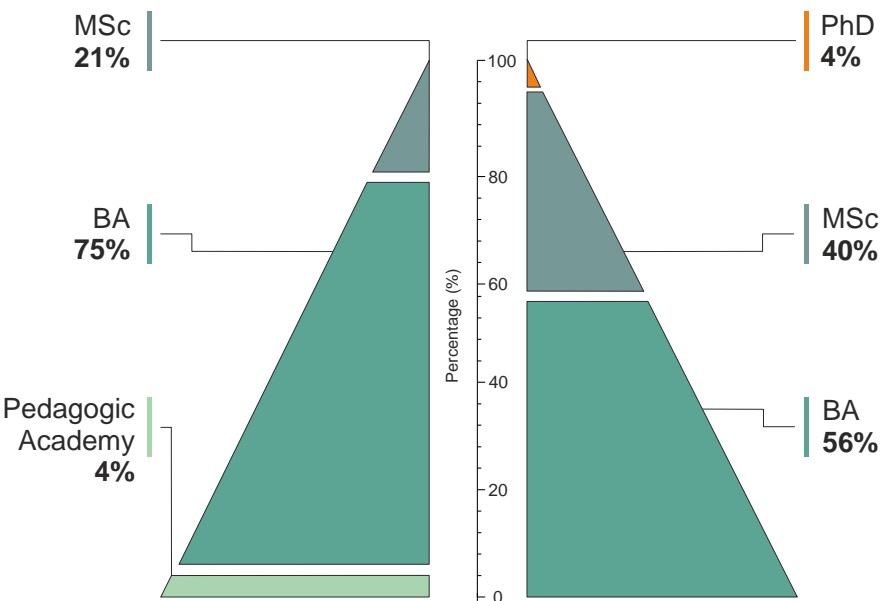

**Figure 3.** Participants' level of studies. Comparison between primary school teachers (left) and LSS teachers (right).

### 3.2. Main Findings

In this section, we present and discuss the main findings of our research.

#### 3.2.1. Didactic Engagement with CCE Topics

At first, the study focused on investigating the level of educators' didactic engagement with CCE topics. As established by their response, 26 (31%) had incorporated CC topics in their teaching as part of some extracurricular environmental education project that they had implemented at some point in their career (see Appendix A.3.). This finding is consistent with results reported by Yanniris [48].

As far as the educational resources used for implementing the above-mentioned projects, analysis of the responses to the follow-up questions revealed that educators preferred the materials found through personal desktop research (the most mentioned portals included BBC Earth, National Geographic, NASA, and ESA), and materials provided by the environmental education coordinators, the Environmental Education Centers in Greece (KPE), and other organizations like museums.

Teachers who had not been didactically engaged in CCE, were asked about the reasons for their abstention. The majority expressed the view that although CC is a very important issue that will greatly impact the future of their students, they do not feel confident enough to teach it. They also added that, as far as extracurricular projects are concerned, there is a wealth of topics to choose from, which are equally important, but easier to teach (ranging from human rights and respect of diversity to road safety and prevention of addictions). No teacher attributed their abstention to any controversy over CC.

#### 3.2.2. Awareness and Prior Use of EO Data in CCE

Subsequently, our study focused on recording the level of teachers' awareness and prior use of EO imagery, videos, simulations, as well as EO-based educational resources, as offered by space agencies and programs. For this purpose, a blend of closed- and open-ended questions, as well as "why" and "how" follow-up questions were addressed.

Regarding the use of satellite imagery and videos in the participants' teaching practice, content analysis revealed that 48% of teachers used Google Maps and Google Earth in

their teaching repeatedly. Out of those teachers, 72% were under the age of 50. NASA satellite images or EO videos and simulations were used by 25% of teachers, who were, again, mostly under the age of 50 (by 80%). As far as ESA imagery and video use, only 14% of teachers used this resource, and out of those, 83% were under the age of 50. Most of the educators who used NASA or ESA videos in class mentioned that the hardest part was the real time translation. They also mentioned that the lack of translated material in Greek discouraged their colleagues from following similar practices. The majority of teachers (52%) had not used any form of EO imagery, video, or simulations prior to the interview, and out of those, 55% were over the age of 50. It was also found that the highest educational qualification for the 87% of those teachers was a BA degree.

Further, focusing on the European space and EO sector, only 10 teachers (12%) stated that they were aware of ESA's educational material, classroom activities, and training opportunities for teachers. Of those, two were primary school teachers and eight LSS teachers. The follow-up discussion with the teachers being aware of ESA's educational resources revealed that only two teachers used any of these resources in their teaching. Specifically, the first was a primary school 6th grade teacher who used the "Earth under the lid—Understanding the greenhouse effect" activity from ESA's primary classroom resources. The activity was used within the frame of an environmental project the teacher implemented in collaboration with the school's English teacher, using the Content and Language Integrated Learning (CLIL) approach (see Appendix A.4.). The educator stressed that the activity was well received by the students and that it kept them engaged and motivated. The teacher also suggested that such activities should be available in Greek if further classroom integration was anticipated. The second teacher was an LSS geologist who implemented parts of the activity "Extracting water from lunar soil – Learning about filtration and distillation" from ESA's secondary classroom resources. Again, the lack of similar classroom activities in the Greek language was stressed by the participant. Subsequent discussion with the teachers who were aware of ESA's educational resources revealed that most of them discovered these resources either through colleagues' social media or through personal desktop research.

Interestingly, most study participants were unaware of the Copernicus EO program of the European Union, its objectives, products, services, and the tools it freely provides. Only three educators were aware of the Copernicus EO program, either though press references during the devastating 2018 Attika wildfires [49], or after visiting science festivals (e.g., Athens Science Festival [50] and Researchers' Night [51]). However, all three teachers argued that Copernicus data, services, and tools were most suitable for scientists and tertiary education, not for formal education.

### 3.2.3. Level of Interest for the Utilization of EO Data and Tools in CCE

At this point, the study attempted to record the participant views on whether, EO-based educational resources, in the form of hands-on student activities and EO tools would help educators stimulate student interest in climate change and help them get the message across more clearly. Thus, by means of a five-point Likert scale, participants were asked to rate the anticipated effectiveness of the EO utilization in CCE.

Before posing this question, and in order for our participants to have a better understanding of how EO can be used for CCE, we demonstrated some of the free spatial visualization tools of the Copernicus program (i.e., EO Browser and Sentinel Playground [30,31]). We encouraged participants to recall any past natural or manmade disaster in their region of interest, and subsequently guided them to locate the area, make a simple query for a specific date range, and browse through the available true color images, in order to demonstrate the intuitive nature of the tools. Depending on the interest and background knowledge of the participants, selected scenes were displayed in different combinations of spectral bands. Furthermore, we presented printed worksheets of the EO-based student activity "Melting ICe Effects" (MICE) [52], which is a translated and locally adapted activity based on ESA's freely available classroom resource "The ice is melting" [53] (see Figure 4).

The participants were encouraged to browse through all sets of the student worksheets and the teacher's guide of this 90-minute classroom activity.

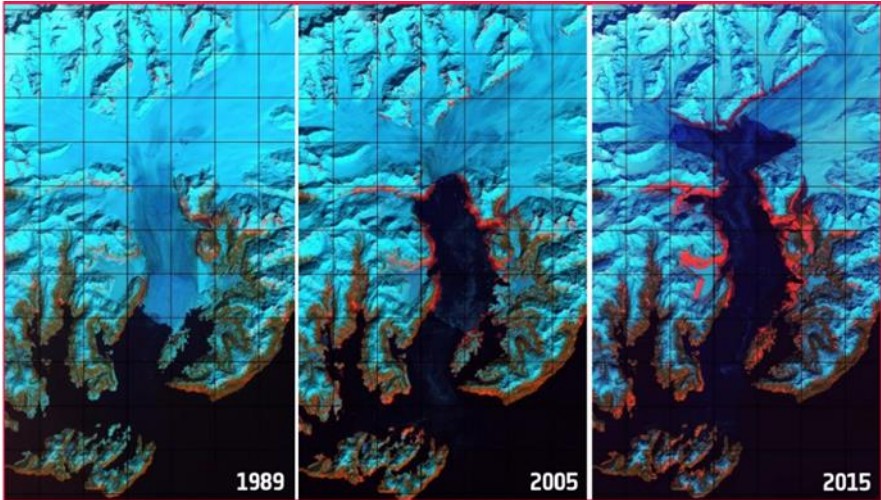

**Figure 4.** Satellite diachronic imagery of the Columbia glacier in Alaska. The image sequence was shown to the teachers to demonstrate the use of remote sensing for the calculation of the glacier's shrinkage during a period of 26 years (taken from the ESA student activity "The ice is melting", which is part of the "Teach with space" collection for the primary school level).

As established by the participant ratings, 51% of teachers expect that EO use would be "Very effective", 30% "Effective", 18% "Fairly effective", and 1% "Slightly effective". Based on the follow-up discussion, it was evident that two major concepts prevailed in teachers' reasoning behind their answers. The majority (81%) who thought that EO use would be either "Very effective" or "Effective" believed that "Anything related to Space attracts student interest" and that "Visualizations and animations of any kind are very effective in getting a message across". On the other hand, teachers who thought that EO use would be either "Fairly effective" or "Slightly effective" believed that the interest to climate change, environmental issues, and space itself, depends mostly on the personality of the student and the awareness within the family environment. "Indifferent students will not be stimulated by such means" they quoted. Our data reveal that the teachers who strongly believed in the effectiveness of EO utilization in CCE were mostly the ones who completed in-service training on ICT. This may imply that higher familiarity with technology may affect teachers' views on the use of EO. No significant relation of the above preferences with the age of the sample or their level of studies was found in our data set.

3.2.4. Preferred Content for Future Educational Material.

At this stage of our study, we aimed to gather the teacher preferences on the types of content and thematic areas that should be covered by future EO-based educational material for CCE. From the discussions in the pre-research phase, we recognized that teachers' responses on the desired thematic areas of interest could be grouped in two major categories:

1.  Local climate change impacts, including forest fires, floods, droughts, etc.
2.  Global climate change impacts and causes, including temperature rise, greenhouse effect, polar ice melting, and sea level rise.

As such, we formed a question on the preferred didactic approach for climate change i.e., either from its global–planetary cause-and-effect perspective, or from the perspective of local impacts, in the form of intense and more frequent natural disasters. Participants were asked to rate their level of interest by means of a five-point Likert scale. As established by the teachers' ratings, local climate change impacts attracted most of the participants' interest.

Specifically, 70% of teachers were "Very interested" and 21% were "Interested" while the remaining 8% were "Fairly interested" in local extreme events. Global climate change impacts and causes also attracted the interest of participants, but to a lesser extent: 50% of teachers were "Very interested", 20% were "Interested", 27% were "Fairly interested", and 2% were "Slightly interested".

As revealed by numbers, local climate change impacts are the preferred thematic for incorporating CC into teaching. This is true for both populations, primary and secondary school teachers. However, a notable difference is observed between primary and lower secondary education level. Interestingly, primary school teachers prefer educational material for local disasters far more than LSS teachers (see also Figure 5).

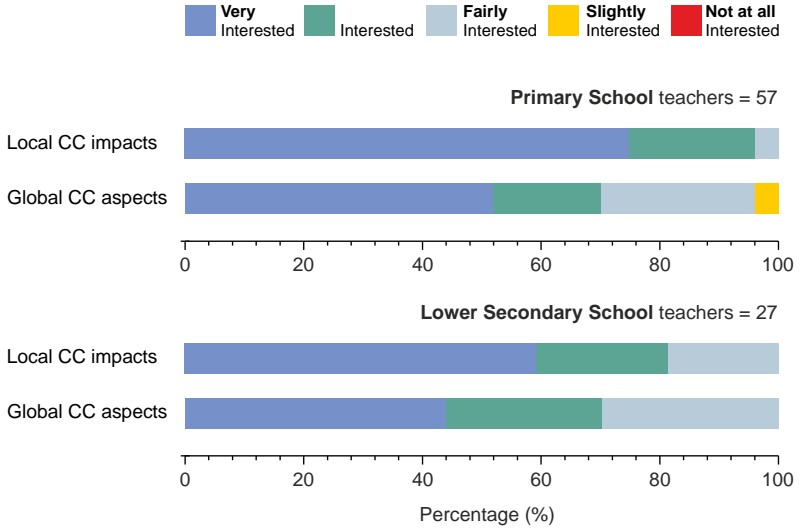

**Figure 5.** Level of interest for local or global aspects of CC between primary school teachers (upper panel) and LSS teachers (lower panel).

During the discussion that followed, primary school teachers expressed the view that their students would respond better to enquiries on environmental issues they may have experienced or are familiar with, compared to more abstract planetary issues. Furthermore, it was stated that primary school students may have difficulties addressing planetary environmental issues, considering that global maps are introduced for the first time in the 6th grade geography lesson.

Subsequently, we collected teacher preferences for environmental phenomena that can be visualized by satellites, but are less popular and indirectly connected to climate change. In the interview guide, we grouped indicative environmental issues in three categories, based on the respective satellite visualization capabilities, as follows:

- Land: forest fire scar regeneration, land use changes.
- Air: desert dust outbreaks, air pollution from anthropogenic activities.
- Ocean: oil spills, ocean garbage patches, phytoplankton growth.

Participants were asked to rate their level of interest on the defined thematics by means of a five-point Likert scale. As established by the educators' ratings (see also Figure 6), for land visualizations, 32% of teachers were "Very interested", 19% "Interested", 38% were "Fairly interested", 8% were "Slightly interested" and 2% "Not at all interested". For Air, 52% of teachers were "Very interested", 24% as "Interested", 21% were "Fairly interested", and 2% were "Slightly interested". For Ocean, 67% of teachers were "Very interested", 25% "Interested", and 8% were "Fairly interested".

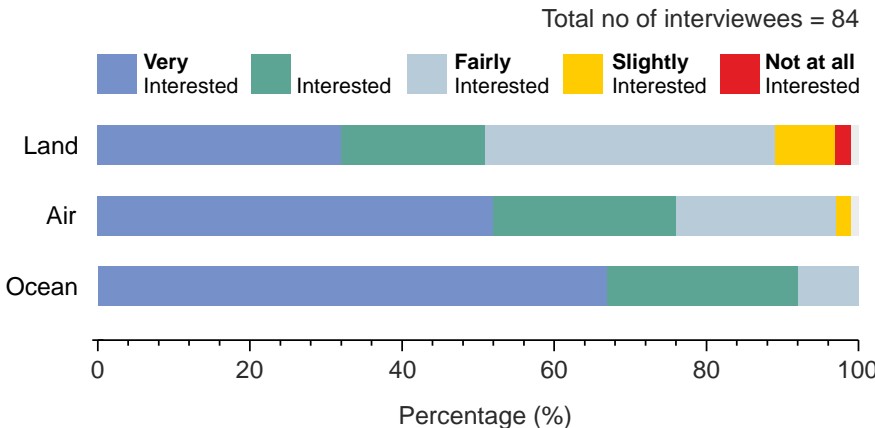

**Figure 6.** Levels of participants' interests for phenomena falling under land, air, and ocean categories of satellite visualizations.

It should be noted that the "Land" category was misinterpreted to include earthquakes on top of the suggested phenomena and, therefore, the category was originally rated with "Very interested". However, when it was clarified that earthquakes are not in any way linked to climate change (which is a common misconception [54,55]), nor can be directly visualized by satellites, the level of interest dropped. For the "Atmosphere" category, desert dust outbreaks gathered most of the interest, despite being a phenomenon with hardly any textbook references in the school curriculum. Teachers pointed out that desert dust is a phenomenon that they are personally interested in. They anticipated that it would attract student curiosity and, therefore, should be addressed in more subjects. The overall high interest for the "Ocean" category is attributed to ocean plastics and oil spills, according to the educators' argumentations. These environmental issues are frequently studied in schools and are in textbook references in many curricular subjects. However, it should be noted that the "phytoplankton growth visualizations" received enthusiastic response only after its connection with oxygen production was explained.

No significant difference on the thematic level of interest was observed between primary and LSS teachers. Both groups rated ocean-related visualization as the most useful for their teaching; this was followed by air and, subsequently, land. In the follow-up discussion, teachers were probed to share ideas for additional satellite-observable environmental issues. Their suggested ideas are grouped in the following categories:

- War zone desertification: detecting war zone devastation from space and comparing pre-war and post-war visualizations including possible desertification of areas depleted by war.
- Rubbish dump: detection of illegal rubbish dumps (mainly proposed by rural schools; non-rural west Attika schools were mostly interested in monitoring the expansion rate of legal rubbish dumps).
- Water supply lakes: monitoring lake level variations.
- Stone quarries: identification of local stone quarries and how the landscape is transformed over time.

From the content analysis of the feedback received, it was revealed that most of the ideas were related to teachers' personal interests for specific local environmental problems or to specific school subjects, such as history, literature, and environmental studies. Surprisingly, most of the ideas were related to non-STEM subject areas. Further, it was also apparent that proposed ideas involved land use changes, although land visualizations came last in teacher preferences at the previous stage.

3.2.5. User Requirements for Future Educational Material Format

Educators were finally asked to discuss the format of their preference for the EO-based CC educational material to be most useful for classroom teaching. Specifically, they were

asked to choose between extended activities for extracurricular project-based use, sort activities that could be fitted in the didactic time of an everyday lesson (also known as "activity-shots"), or both. The "Extended activities" were defined as activities that may require the class to perform experiments, fill in worksheets, watch videos, and simulations, produce satellite imagery animations showing progression of events over time, record measurements, etc. Such activities require a minimum of 90 minutes or more of lesson time, significant teacher preparation, and the use of equipment that is not already available in the classroom. On the other hand, activity-shots were defined as activities with very little or no teacher preparation, use of equipment that is already available in the classroom, and a duration that does not require more than 10–20 minutes of lesson time.

Based on the participants' responses, the majority, 64% of teachers, chose activity-shots as the preferred and most useful format. They argued that this type of activity is more likely to be incorporated into their lessons, especially if it is directly linked to specific curricular topics. The teachers also pointed out that it would be unlikely to find extra lesson time required to implement an extended activity unless they implemented a specific extracurricular project. They complained that curriculum pressure does not leave them with enough free classroom time, as similarly stated in other studies [56,57]. Moreover, 15% of teachers chose extended activities and 20% preferred that both formats be available. This group of teachers argued that extended activities are far more comprehensive and allow them to extract small segments for use during their everyday lessons if they so desire. They further argued that EO-based extended activities can be especially useful for educators, implementing optional environmental projects, or any other extracurricular program, but also for educators using blended teaching methods, such as the flipped learning approach (where didactic time pressure is not an issue, since students gain their first exposure to new content outside of class, and then use classroom time to discuss, challenge, and apply the new knowledge [58]).

## 4. Discussion and Conclusions

Our study on CCE in Greece revealed that only 31% of the participating educators incorporated climate change topics in their teaching, mostly through extracurricular projects. In order to increase climate literacy in Greece, we examined the case of adopting modern methods and good practices applied in other countries for promoting CCE in schools. We specifically inquired the teachers' perspectives on the utilization of EO for facilitating CCE in a modern way. EO has been widely recognized as an innovative teaching and learning approach for CCE, as it illustrates climate change impacts most effectively with a high degree of vividness and is capable of bringing significant benefits to formal education. Modern satellite RS technologies provide impressive illustrations of earth's climate system, and its changes allowing for spatio-temporal monitoring of the state of the atmosphere, land, and oceans. Highlights include important discoveries about the climate system that have not been detected by climate models and conventional observations; for example, the spatial pattern of sea-level rise, and the cooling effects of increased stratospheric aerosols. New insights are made feasible by the unparalleled global visualizations, which have the capacity to engage students and increase their motivation.

This idea was welcomed and embraced by the teachers of our study, who were initially unaware of the EO data and tools (on a percentage of the order of 90%) and considered RS for EO highly specialized and of a technical nature. Nevertheless, when the study participants were introduced to examples of EO educational resources and tools, they recognized the potential of EO as a very effective approach for attracting students' interest and engaging them in active learning.

Subsequently, for the future development of EO-based educational material for CCE, the teachers of our study favored the development of simple, short activities (activity-shots), where they could fit into the didactic time of a classroom lesson with minimal preparation effort. Regarding the thematic content of the activities, they suggested that local climate change impacts should be given priority over global climate change phenomena and

that such activities should be directly linked to topics in the existing curriculum. From the discussions followed, we conclude that this simple form of EO-based educational material for CCE is not only considered helpful for increasing climate literacy, but also important for boosting EO penetration in schools by convincing more teachers that RS is not "rocket science". Regarding the alternative form of classroom activities (i.e., the extended EO-based activities), teachers argued that these would be especially useful, either to educators implementing optional environmental projects or to educators using a blended teaching methodology, such as the flipped classroom approach. According to the literature, the combination of flipped learning and remote sensing is considered effective for environmental education [59] and, thus, extended EO-based activities should be offered along with the preferred activity-shots.

Teachers have also suggested that climate-related phenomena, observable from space, should not be associated only with STEM curricular subjects, but also with non-STEM subjects, such as history, social studies, literature, art, etc. Therefore, our analysis indicates that teachers would welcome more initiatives (e.g., "Climate Across Curriculum project by Smith College" [60]) that offer ready-to-use educational material that combines EO-based CCE topics and non-STEM curricular subjects.

In order to promote climate literacy in schools through EO and in order to improve teachers' interests in the topic, teacher-training initiatives should be adopted by the respective education authorities, academia, or space agencies at a national level. Meanwhile, EO for CCE could be promoted through existing well-established national or pan-European teacher training structures, which offer a wealth of online courses, webinars, tutorials, and teaching materials to thousands of teachers across Europe [61–64]. These structures maintain networks of local ambassadors who support local curricular and language needs. By training expert teachers from within these structures, awareness, interest, and use of EO for CCE could be amplified.

For Greece, in particular, additional future work on the development of appropriate EO-based CC educational material in the native language is necessary. The authors aim to work towards this direction, building on previous efforts already taken (see Appendix B). We envision that the existing educational material, as well as new material that will be developed in the future, according to the needs, preferences, and ideas collected in this study, will be made available through a dedicated educational portal, currently under construction. However, all of these efforts will remain fragmented without an effective coordination at a national level. The recent establishment of the Hellenic Space Center [37] in Greece may pave the way for the establishment of an ESA-ESERO office that could tackle many of the limitations discussed here.

**Author Contributions:** Conceptualization, methodology, formal analysis, investigation, and writing—original draft preparation, P.A.; data curation, M.H.; resources, writing—review and editing, supervision, P.N., M.H., E.V., A.A. All authors have read and agreed to the published version of the manuscript.

**Funding:** The APC was funded by the project "PANhellenic infrastructure for Atmospheric Composition and climate change" (MIS 5021516), which is implemented under the action "Reinforcement of the Research and Innovation Infrastructure", funded by the operational program "Competitiveness, Entrepreneurship and Innovation" (NSRF 2014–2020) and co-financed by Greece and the European Union (European Regional Development Fund).

**Data Availability Statement:** The data supporting results can be found in the following Zenodo link: http://doi.org/10.5281/zenodo.4699199 (accessed on 17 March 2021) (Reference: Panagiota Asimakopoulou. (2021). Dataset for Article in MDPI Remote Sensing Journal -Manuscript ID: remotesensing-1168493 (Version 1.0))

**Acknowledgments:** P.A. is grateful for the freely available EO-based classroom activities provided by ESA. P.A. thanks ESA for offering the teacher training workshop. The authors also thank the Copernicus Academy for the guidance and the opportunity to connect with other active members in EO education, as well as the provision of Copernicus dissemination material and goodies. P.A. gratefully acknowledges the personnel of the three (3) museums of the School of Science of NKUA, namely the Mineralogy–Petrology Museum, the Paleontology Museum, and the Zoological Museum. This research would not be accomplished without their support and hospitality.

**Conflicts of Interest:** The authors declare no conflict of interest.

## Appendix A

### Appendix A.1. Greek Teachers' Initial Education

Up until 1984, prospective teachers of primary education were trained only in the Pedagogic Academies, public institutions, which offered a 2-year lower-level higher education qualification. As of 1984, prospective primary school teachers could receive a bachelor's degree by the newly established Pedagogical Departments for primary education in Greece's public universities. The two systems co-existed until 1988, when the Pedagogic Academies were abolished. Consequently, from 1989 onwards, there have been two pools of primary teachers eligible for recruitment [65]. During the subsequent years, the state offered in-service primary school teachers, holding a Pedagogic Academy diploma, the opportunity to upgrade them to four-year university level degrees, by implementing country-wide, in-service teacher equalization program, called the Exomiosi program, between the years 1997 and s1999 [66]. In our study, it is worth mentioning that 23 out of 25 teachers holding a Pedagogic Academy diploma had completed an Exomiosi program and, therefore, upgraded their degree.

### Appendix A.2. In-Service Teacher Training Program on the Use of ICT

The first phase of the in-service teacher training program in the use of ICTs in education started as early as 2002 and it was subsidized by EU funds [65]. During its second phase (2014–2020), the program was renamed (In-Service Training of Teachers in the utilization and application of digital technologies in the teaching practice) and restructured. The earlier A-level training for basic ICT skills became B1-level and the old B-level became new B2-level advanced training for the use of ICT in classroom teaching [67].

### Appendix A.3. Types of Environmental Education Projects Implemented in Greek Schools

In Greek schools up to now, CCE is optionally implemented by teachers, through project-based environmental activities like the following:

- Optional school activities. These are extracurricular activities teachers can undertake at the beginning of the school year (they can choose between actions on Environmental education, Health education or Culture and arts-oriented topics) [68].
- Programs implemented by a nationwide network of local Environmental Education Centers—KPE
- eTwinning programs. These are extracurricular projects designed and implemented collaboratively by groups of partner schools from the eTwinning community. eTwinning is a free online community for schools in Europe and some neighboring countries, which allows schools to find partners and collaborate on projects within a secure network and platform [69].
- UNESCO's Associated Schools network—ASPnet projects

In 2020, as part of the national reforms in school education, Greece introduced the skills workshops [70]. These are foreseen as compulsory school activities covering a wide variety of thematics including climate change.

*Appendix A.4. The Content and Language Integrated Learning (CLIL) Approach*

CLIL is an approach to foreign language teaching, where students learn a foreign language through the content of different curricular subjects such as geography, fine arts, science etc. In this approach, language teachers and subject teachers work together teaching both the subject and the language at the same time. It is based on the principle that foreign language learning is improved when students are interested in a topic which motivates them to acquire the required language to communicate [71,72].

**Appendix B**

Actions towards the development of EO-based CCE educational activities for Greek schools have been taken during and after the period of the current study by the authors. Specifically, we proceeded with: (a) the pilot-testing of the MICE classroom activity [52], (b) the development and pilot-testing of a new EO-based classroom activity (MaFiS) [73] build around a major local disaster of the Attika region, (c) the translation of educational videos on atmospheric hazards experienced in Greece like desert dust [74], and (d) the production/creation of TV lessons on Greek state TV which promote the use of EO tools for CCE, based on curricular 6th grade Geography topics such as Atmosphere-Greenhouse Effect [75], Natural Disasters [76], and Deserts [77].

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
