# Peer review of "Earth Observation as a Facilitator of Climate Change Education in Schools: The Teachers’ Perspectives"

_remotesensing, doi:10.3390/rs13081587_

Round 1

Reviewer 1 Report

Good paper. Remote sensing community indeed needs to spread knowledge and data to a broad audience. Last ten years we experienced many extreme climate events.  

Author Response

We would like to thank the reviewer for his/her positive evaluation.

Reviewer 2 Report

Dear all,

Thank you for the opportunity to read your paper.

I think the work is well structured and the aim is scientifically useful. However, I think that the results should be discussed more widely, trying to increase the scientific approach of the work. Perhaps the authors should find and suggest solutions to improve teachers' interest in the topic of climate change and offer tools to increase interest.

The below aspects should be addressed by the author before publication.

SPECIFIC COMMENTS:

- Keywords: “primary school education; secondary school education”, I’d use only “school education”.

- Line 293 to 297: I think it might be interesting to see if there is a certain link between educational qualification, age and teaching approach used as well as interest shown (line 350)

Reviewer 3 Report

The article deals with the issue of climate change in the school education system, and the availability of remote sensing material as tools to increase change. The article is well written. The research methods are clear and understandable. The results are also well understood. The authors of the article did not write at all about the causes and effects of climate transformation with regard to geological time. The article is written in a journalistic way, the reader feels that the climate changes are only the result of anthropopressure. There is no broader context for embedding research here.  In principle, a scientific article should provoke reflection by suggesting materials for discussion. As many researchers have shown, climate change is a phenomenon known in history of our planet. In the geological history of a given area, such as Greece, many times the climate was much warmer than it is today. It is worth presenting, for example, the works of Milancovic. (https://en.wikipedia.org/wiki/Milankovitch_cycles).

Human activity is a factor accelerating, not causing, climate change. Natural geo and cosmogenic processes are the main factors of climate change. The authors omitted this aspect altogether.  Teachers as part of the public education system, first of all aware of the genesis of climate change phenomena, to improve the competent and reliable communication of information on climate change in a broader sense.

Please include this information in the introduction and in the summary and discussion section. After adding this data, I would like to review the manuscript again.

Best Regards

Author Response

We thank the reviewer for the positive evaluation of our work and the specific scientific comments. We consider however that the reviewer’s suggestions are beyond the scope of our paper for the following specific reasons:

  • In our work we focus on the human impact on climate change during the last decades, so as the students will gain knowledge on their potential climate footprint and on adaptation and mitigation measures needed to avoid future climate change impacts. This principle has been set by the UNESCO’s guidelines for CCE and sustainable development (already referenced in the text as the basis of our approach), stating that: “Climate Change Education is a powerful tool that prepares young generations to live with the impacts of climate change and empowers learners to take appropriate actions to adopt more sustainable lifestyles”.
  • Our work demonstrates the added value of using EO in schools for facilitating CCE. We assume that the envisioned content of future EO-based educational material for CCE would focus on remote sensing visualizations and diachronic trends showing the human impact on climate, from the anthropogenic greenhouse emissions to global warming. This 50-year time span of remote sensing era is the Anthropocene, a period that cannot reveal geological and paleoclimatic patterns from the specific EO datasets.

Round 2

Reviewer 3 Report

I understand this approach to the subject.

Nevertheless, I would not like my children to learn from these teachers after these courses.